# Prevalence and factors associated with antenatal care service access among Indigenous women in the Chittagong Hill Tracts, Bangladesh: A cross-sectional study

Shahinoor Akter[1,2,3,4]*, Jane Louise Rich[1,5], Kate Davies[6], Kerry Jill Inder[2,3,7]

1 School of Medicine and Public Health, Faculty of Health and Medicine, University of Newcastle, Callaghan, New South Wales, Australia, 2 Priority Research Centre for Generational Health and Ageing, University of Newcastle, Callaghan, New South Wales, Australia, 3 Hunter Medical Research Institute, New Lambton, New South Wales, Australia, 4 Department of Anthropology, Jagannath University, Dhaka, Bangladesh, 5 Centre for Brain and Mental Health Research, Callaghan, New South Wales, Australia, 6 School of Humanities and Social Science, Faculty of Education and Arts, University of Newcastle, Callaghan, New South Wales, Australia, 7 School of Nursing and Midwifery, Faculty of Health and Medicine, University of Newcastle, Callaghan, New South Wales, Australia

* Shahinoor.Akter@uon.edu.au

**Data Availability Statement:** Following ethical approval provided by the Human Research Ethics

## Abstract

### Background

Prevalence of accessing antenatal care (ANC) services among Indigenous women in the Chittagong Hill Tracts (CHT) is unknown. This study aims to estimate the prevalence of accessing ANC services by Indigenous women in the CHT and identify factors associated with knowledge of, and attendance at, ANC services.

### Methods

Using a cross-sectional design three Indigenous groups in Khagrachari district, CHT, Bangladesh were surveyed between September 2017 and February 2018. Indigenous women within 36 months of delivery were asked about attending ANC services and the number who attended was used to estimate prevalence. Socio-demographic and obstetric characteristics were used to determine factors associated with knowledge and attendance using multivariable logistic regression techniques adjusted for clustering by village; results are presented as odds ratios (OR), adjusted OR, and 95% confidence intervals (CI).

### Results

Of 494 indigenous women who met the inclusion criteria in two upazilas, 438 participated (89% response rate) in the study, 75% were aged 16–29 years. Sixty-nine percent were aware of ANC services and the prevalence of attending ANC services was 53% (n = 232, 95%CI 0.48–0.58). Half (52%; n = 121) attended private facilities. Independent factors associated with knowledge about ANC were age ≥30 years (OR 2.2, 95%CI 1.1–4.6), monthly household income greater than 20,000 Bangladeshi Taka (OR 3.4, 95%CI 1.4–8.6);

Committee of the University of Newcastle, Australia, non-identifiable data may be shared with other parties to encourage scientific scrutiny and to contribute to further research and public knowledge. A data file converted to PDF format is attached as a supplementary file. Data can also be shared upon request via emailing NOVA, the University of Newcastle open access repository on Nova@newcastle.edu.au.

**Funding:** The authors received no specific funding for this work.

**Competing interests:** The authors have declared that no competing interests exist.

knowledge of pregnancy-related complications (OR 3.6, 95%CI 1.6–8.1), knowledge about nearest health facilities (OR 4.3, 95%CI 2.1–8.8); and attending secondary school or above (OR 4.8, 95%CI 2.1–11). Independent factors associated with attending ANC services were having prior knowledge of ANC benefits (OR 7.7, 95%CI 3.6–16), Indigenous women residing in Khagrachhari Sadar subdistrict (OR 6.5, 95%CI 1.7–25); and monthly household income of 20,000 Bangladeshi Taka or above (OR 2.8, 95%CI 1.1–7.4).

## Conclusion

Approximately half of Indigenous women from Chittagong Hill Tracts Bangladesh attended ANC services at least once. Better awareness and education may improve ANC attendance for Indigenous women. Cultural factors influencing attendance need to be explored.

## Introduction

Maternal health outcomes have been the center of public health-related international manifestos since the Millennium Development Goals were signed by 189 world leaders in 2000 [1]. Globally, the maternal mortality ratio has declined by 75% between 1990 and 2015 [1, 2]. Despite being a lower-income country, Bangladesh reduced its maternal mortality ratio by 40% from 322 (95%CI 259–391) to 194 (95%CI 149–238) deaths per 100,000 live-births between 2001 and 2010 [3]. A significant increase in the use of maternal health care (MHC) services including antenatal care (ANC) service attendance is a key contributing factor to reducing maternal mortality [4, 5].

The proportion of women in Bangladesh receiving at least one ANC visit from a medically trained health care provider increased from 28% in the early 1990s to 64% in 2014 [4, 6]. However, this has not meant that the reduction has been inclusive and equitable for women from different socio-cultural backgrounds [5]; rather there is a lack of information regarding MHC accessibility and utilization including ANC visits for cultural minority groups including Indigenous women in the Chittagong Hill Tracts (CHT) [7, 8].

Antenatal care is important for risk identification, prevention, and management of pregnancy-related complications during pregnancy [9, 10]. Antenatal care provides opportunities to educate women about reproductive health including safe delivery and postnatal care services [10, 11]. Timely and routine ANC from skilled health care providers is a key determinant of improved health outcomes for mother and child during pregnancy [6, 9, 10]. During the Millennium Development Goal era, the World Health Organization (WHO) recommended that a pregnant woman should have her first ANC check within the first trimester of pregnancy, and should complete at least four ANC visits during pregnancy [10]. In 2016 WHO modified the recommendation to a minimum of eight ANC visits as part of the Sustainable Development Goals [10, 12].

Chittagong Hill Tracts is home to eleven Indigenous communities consisting of three districts–Bandarban, Khagrachhari and Rangamati—located in south-eastern Bangladesh [13–15]. Since the colonial period, CHT has been in conflict and isolated politically and socially, even after Independence in 1971. A Peace Accord was signed in 1997 between the Bangladesh Government and Indigenous communities to stop the long history of political tensions [14], however, like other countries around the world, CHT Indigenous people have received minimal policy recognition to ensure their human rights, including health rights. Data on

Indigenous people's health including women's health has not been well documented in national demographic surveys, even after twenty years of singing the Peace Accord [16].

Antenatal care service utilization among CHT Indigenous women has been reported in the Mru Indigenous group from the remotest part of the Bandarban hill district, Bangladesh indicating that place of residence, age, school attendance, distance to health service centres and access to mass media were associated with lower ANC utilization [17]. Data from a multi-sectoral capacity building and development program for all CHT inhabitants [18] revealed that the program boosted ANC-checks (18%) among Indigenous and non-Indigenous women with experience of violence in the intervention areas [18].

The Human Development Research Centre (HDRC) baseline survey in 2009 [13] found that ANC checks were lower among CHT Indigenous women (34%) than their Bengali counterparts (52%). Moreover, Indigenous women lacked knowledge of basic MHC services [19]. No further evidence was found on CHT Indigenous women's health issues to assess or meet their needs [16, 20], indicating that Indigenous women have not been prioritised for intervention programs to improve health outcomes despite having the worst health record in the country [7, 8, 21]. Evidence suggests that maternal anthropometric and socio-economic characteristics, including ethnicity, have a significant association with the birthweight of newborn babies [22]. The percentage of low-birth-weight newborn babies was found to be highest in CHT Indigenous women compared to non-Indigenous women in a report published by UNICEF 2015 (United Nations Children's Funds) [8, 23].

Evidence from lower and middle-income countries (LMICs) such as China and India shows that lower levels of school attendance, lack of awareness of ANC services, involvement in income-generating activities, limited doorstep services, and cost are key barriers to Indigenous women accessing ANC services [24–27]. A recent systematic review indicated that health intervention programs for Indigenous women in LMICs where community-engagement was not prioritised as part of the development and implementation of the intervention resulted in culturally unfriendly MHC services [28]. This may impact access to ANC. Given the socio-political context and scarce evidence on CHT Indigenous women, it is impossible to draw the same conclusion for this particular group. This study attempts to provide evidence about the prevalence of attending ANC services and factors associated with knowledge and attendance at ANC services among CHT Indigenous women in Bangladesh. Findings have the potential to help policymakers create a more inclusive and universal public health strategy to improve ANC access and help achieve the WHO Sustainable Development Goals.

## Method

### Study design

The study used a cross-sectional survey design. Reporting is guided by the STROBE statement for cross-sectional studies [29].

### Study setting

This survey was conducted in two subdistricts/upazilas (Matiranga and Khagrachhari Sadar) of Khagrachhari hill district between September 2017 and February 2018; full details have been previously described [30]. The study site was purposively selected due to available data on Indigenous population size and health facilities. In Khagrachhari hill district, primary and secondary level health care facilities plus private facilities provided MHC services at the time of data collection as shown in Table 1. Each subdistrict has one health complex (Secondary level care). The only private clinic was located in Khagrachhari Sadar upazila (subdistrict). Each upazila has Union Health Centres and Family Planning Centres. Of 26 Union Health Centres,

**Table 1. Available health care facilities at Khagrachhari Sadar and Matiranga upazial in Khagrachhari hill district (source: District statistics 2011: Khagachhari [31].**

| Level of care | Types of health facilities | Khagrachhari Sadar upazila* | Matiranga upazila* | In Khagrachhari district |
|---|---|---|---|---|
| Secondary level | Health Complex | 1 | 1 | 8 |
| Primary level | Union health centre | 5 | 8 | 26 |
| | Community clinic | 2 | 9 | 47 |
| Private care | Private hospital/clinic | 1 | 0 | 1 |
| | Diagnostic centres | 4 | 3 | 21 |
| Charitable | Missionary hospital and charitable dispensary | 0 | 2 | 2 |

*Upazila = subdistrict.

five are in Khagrachhari Sadar upazila and eight are located in Matiranga upazila. Of the 21 private diagnostic centres in Khagrachhari, four were situated in Khagrachhari Sadar and three were based in Matiranga [31]. At the time of data collection, one Non-government Organisation satellite clinic provided door-step MHC services in limited areas of Khagrachhari district.

## Participants

Indigenous women aged 15 to 49 years who resided in their *para* (village) for at least six months prior to survey completion were eligible to participate. Women within 36 months of delivery were included to allow comparison with data from the Bangladesh demographic health survey [31] and to minimise recall bias about service access. Non-Indigenous women living in the same *paras* were excluded.

## Variables

Independent variables collected included socio-demographic characteristics (age, ethnicity, religion, education, occupation, their partner's education and occupation, household income and place of residence), reproductive history (age at first pregnancy, number of pregnancies, negative pregnancy outcome (miscarriage/ stillbirth), experience of pregnancy complications), knowledge about any nearest health facilities, distance to facilities, knowledge of pregnancy-related complications, knowledge about ANC benefits and access to media (television, radio, mobile and newspaper).

Primary outcome variables were the proportion of women with knowledge about ANC services and the estimated prevalence of accessing ANC services. Participants were asked if they had prior knowledge about ANC (Yes or No), and if so were asked if they accessed ANC services during their last pregnancy (Yes or No).

Secondary outcomes were key sources of information about ANC services, type of facility where ANC services were accessed, reasons for accessing ANC services, number of total ANC visits made, payment for ANC services, language (mainstream or Indigenous language) used during consultation, health education received during ANC check and factors associated with knowledge and access to ANC services.

Where variable categories contained less than 5 responses, categories were combined to facilitate statistical regression. This occurred for participants education status their partners' education status, and for participants' and partners' occupation status. The variable "number of pregnancies till survey date" was categorised as: '1–2 pregnancies' and '3 and above' to reflect the Bangladesh Population Policy that encouraged "No more than two children, but one is better" for all married couples [32]. Pregnancy-related complications were categorized

as mild, moderate (pregnant women should see a doctor) or severe (needs immediate medical intervention) according to the severity of complications using WHO and United Nations Children's Fund recommendations [33].

## Data sources and measures

The data collection tool was a survey based on the Bangladesh Demographic Health Survey questionnaires [34]. Questions on ethnicity and knowledge on MHC and available health facilities were added to the survey which was pre-tested by two female Indigenous field assistants prior to survey administration. The survey was conducted in three Indigenous communities: Chakma, Marma, and Tripura informed by the Karbari (community leaders) and local community health workers. The questionnaire was translated from English into Bangla by a local translator, and cross-verified using independent speakers of Bangla and English.

Using the survey tool data collected from eligible Indigenous women predominantly related to self-reported knowledge about and access to ANC services at primary health care facilities including community-based skilled health care workers; socio-demographic characteristics; reproductive history including age during first pregnancy, number of pregnancies till date when the interview took place, outcome of pregnancies and expected number of children; knowledge about nearest available services for women of reproductive age and utilisation of these services. The survey included questions on distance between home and the nearest health facilities and the research team also estimated this distance using a mobile Global Positioning System (GPS) application.

Before commencing data collection, list of Indigenous *paras* (villages) were collected by contacting local administrative offices and Indigenous community leaders such as Headman (mouza Head) and *Karbaris* (para Head) [13]. List of households with eligible women were collected and reached with the direct assistance of local Indigenous leaders and community-based health workers. The first author along with two Indigenous field assistants administered the survey. The first author was a female non-Indigenous Bangladeshi researcher with an academic degree in Anthropology. The first author grew up in the region where the study was undertaken and has significant work and research experience with Indigenous women of the CHT, and with Indigenous populations globally. Interviews were conducted in the national language, Bangla.

Antenatal care service access data was self-reported and analysed based on the WHO's recommendation of a minimum of four ANC-visits during pregnancy [10]. Indigenous women who attended ANC services at any point during their last pregnancy were considered as having attended an ANC service. Attendance could be at a community-based health facility (community clinic, satellite clinic, school-based clinic, mother or child welfare clinic), a government hospital (district hospital or upazila health complex) or at a private facility. Attendance could also be from a skilled community health worker in their home including from a family welfare visitor, a community skilled birth attendant, or a health worker.

## Study size

Sample size was estimated using a formula for a single population proportion with the assumptions of 95% confidence level (CI), 5% margin of error and an expected 35.2% prevalence of women giving birth in a health institution in the Chittagong Division, Bangladesh [34]. Based on a non-response rate of 20% and the distribution of the three ethnic populations, the survey required 421 Indigenous women: 216 Chakma, 90 Marma, and 115 Tripura women.

## Statistical methods

Software package STATA version 15 (StataCorp LLC, USA) was used to analyse data. All variables were checked for implausible values, errors and missing data. Knowledge of ANC was estimated with 95% confidence intervals (CIs). The numerator was number of women within 36 months of post-delivery who reported knowing about ANC during their last pregnancy. The denominator was the total number of women within 36 months of post-delivery in the two sub-districts.

Likewise, prevalence (95% CIs) of accessing ANC was calculated by dividing the number of women within 36 months of delivery who reported accessing at least one ANC service during their last pregnancy, by the number of women within 36 months of delivery in the two sub-districts.

Secondary outcomes were factors associated with knowledge of ANC services in participating women and factors associated with attendance at ANC services in the sample of women who attended at least one ANC service. To identify potentially important factors associated with these secondary outcome variables with a p-value ≤0.250 following the chi-squared test were purposively included in a multivariable logistic regression model using a backward stepwise approach [35]. The final model adjusted for clustering by *para* (village) reported variables with a p-value <0.100, to avoid excluding potentially important results [35]. Results are reported as odds ratios (OR), adjusted odds ratios (AOR) and 95% CIs.

## Ethical considerations

Although the Indigenous population in Khagrachhari district could speak and understand mainstream Bengali language, the target population consisted of many people who were illiterate or had low-literacy levels [13]]. Therefore, the participant information statement was read out and explained by the study team in Bengali and all aspects of the study procedures were explained to the participants to obtain verbal informed consent. When needed, the information was explained in the participant's local dialect. A literate impartial witness was present during consent and helped to explain the study objectives to the study participants. Participants were given ample time to consult their family members and were able to ask questions before they provided consent for participation. Verbal informed consent was collected prior to survey interview for each participant except for those aged under 18 years. For participants aged less than 18 written or verbal consent from the participant's guardian was collected (husband/ parents or parents-in law) and then informed verbal consent of the participant was secured. Ethical approval, including permission to obtain verbal consent, was obtained from the Human Research Ethics Committee of the University of Newcastle, Australia (H-2017-0204) and the Ethics Committee of the Department of Anthropology at Jagannath University. Participation was voluntary, and privacy and confidentiality were maintained. Data were collected and reported anonymously to ensure privacy and confidentiality.

## Results

Community health workers and *Karbari* reported 494 Indigenous women who were permanent residents and living for the last six months in 47 *paras* in Matiranga (n = 41) and Khagrachari sadar (n = 6) upazila, delivered at least one child within three years preceding the survey. Of these, 51 Indigenous women (11%) were not at home and documented as "missing" after three attempts to administer the survey. The remaining 443 women were invited to participate and 438 provided verbal consent. Five women who consented withdrew from the study before completing the survey. The survey response rate was 89% (438/494) including 220 (50%) Chakma, 100 (23%) Marma and 118 (27%) Tripura women.

## Sociodemographic characteristics of the sample

The mean age of the sample was 25 years (standard deviation ±5.4); most participants (75%) were aged 16–29 years. No participants were aged under 16 years. Most participants resided in the Matiranga (n = 325, 74%) and the remaining in Khagrachari (n = 113, 26%). The majority of participants were Buddhist (n = 326, 74%); did not attend or attended upto primary school (n = 205; 47%); and were from lower income group with a monthly income of 4000–9000 Bangladeshi Taka (equivalent to USD $48-$107, n = 198; 45%). Regarding occupation, half of participants were involved in income-generational work (n = 224; 51%).

## Knowledge and prevalence of access to available ANC services

Of 438 participants, 304 (69.4%, 95%CI 64–74%) reported prior knowledge of ANC services. Prevalence of accessing ANC services was estimated at 53% (232/438; 95% CI 48 to 58%).

## ANC service characteristics

Health care providers at health service centres and the community-based health workers were the primary source of information for participants (n = 168; 55%) regarding ANC services. Of those women with prior knowledge of ANC services 76% accessed ANC services at least once (232/304) and 91 women (39%) attended at least four ANC-visits. Of the women who attended at least four ANC-visits, 54% (n = 49) received information about ANC services from health care providers and the remaining received information mostly from relatives, who were either health care providers or who provided their services during the pregnancy.

Of participants who attended ANC services during their last pregnancy, 121 (52%) attended private facilities and 93 (21%) attended community-based facilities, including district hospitals close to their homes and 18 (4.1%) attended a secondary hospital. A further 57 (24.6%) received their ANC-check at home from trained community health workers such as community skilled birth attendants, family welfare visitors or health workers. Among participants who attended ANC services, 50% paid money for their ANC check. Nearly all (n = 215; 93%) had the consultation in the mainstream Bangla language, however 118 (59%) participants reported they did not understand the information given by the health care providers. Of those who attended ANC services during their last pregnancy, 85 participants (36.6%) attended ANC services in their 1st trimester. Two-thirds (68%) of these women attended ANC services during their last pregnancy due to pregnancy-related complications. Of those who attended ANC during their last pregnancy, 52 (27%) and 32 (22%) attended their first ANC visit to confirm pregnancy and due to previous pregnancy-related complications respectively (see Fig 1).

In response to questions related to health education received during ANC checks,191 (82%) of participants reported their consultation did not cover information regarding pregnancy-related complications.

## Factors associated with knowing about ANC services

Demographic and obstetric characteristics of CHT Indigenous women who had prior knowledge of ANC services compared to those who were not aware of ANC services are shown in Table 2. A significantly higher proportion of Indigenous women who were aware of ANC were younger, had higher school attendance, had a partner involved in service (public services/working at any private organisations/ companies) and business, had a higher household income, had prior knowledge about nearest health facilities, accessed media for MHC, became pregnant before their 20th birthday, had a negative pregnancy outcome and had previous knowledge about pregnancy-related complications during their last pregnancy compared to

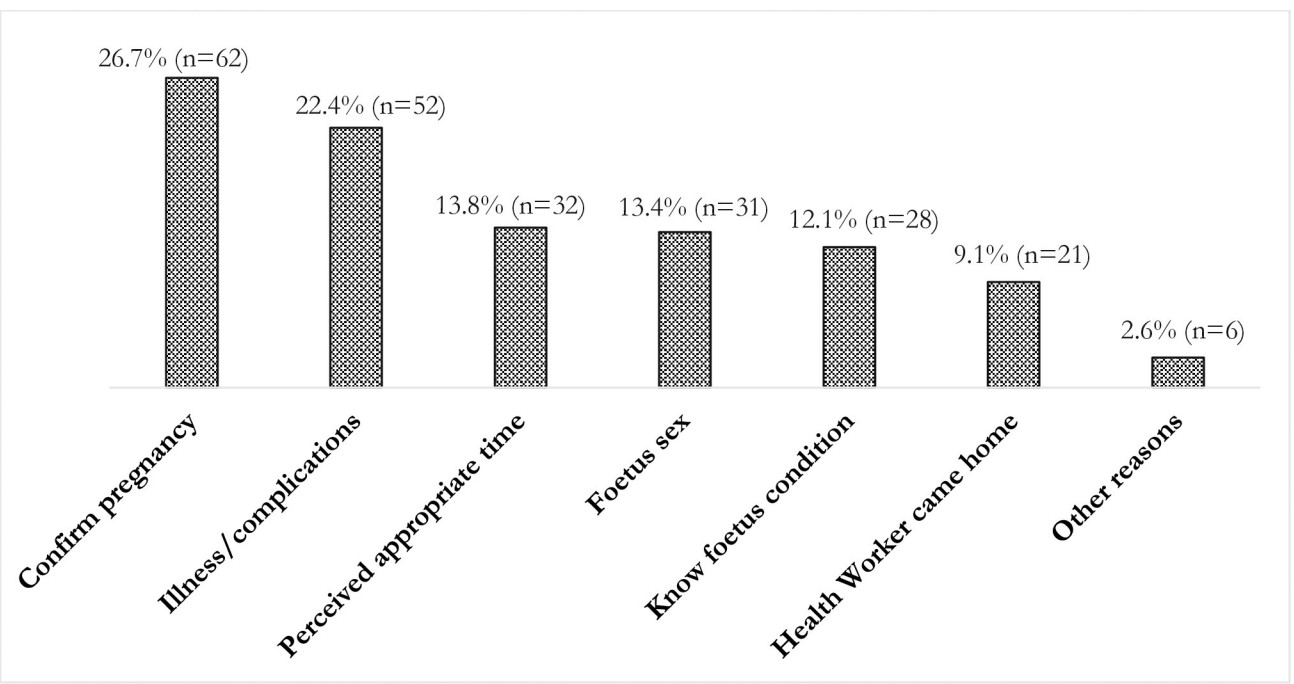

**Fig 1. Reasons for attending 1st ANC check during last pregnancy.**

those who did not know about ANC services. These variables, significant on univariate analysis, were included in the multivariable logistic regression analysis.

In the multivariable logistic regression analysis, five factors were independently associated with prior ANC knowledge. Indigenous women aged 30 and above had twice the odds (AOR 2.2, 95% CI 1.1–4.6) of having prior knowledge about ANC services compared to those aged between 16 and 29 years. Indigenous women who attended secondary school and above had nearly five times the odds of having prior ANC service knowledge (AOR 4.7, 95% CI 2.0–11.1) compared to Indigenous women who attended up to junior school. Compared to Indigenous women with a monthly household income between 4000 and 9000 Bangladeshi Taka (BDT; or USD $47.6-$107), Indigenous women with a monthly household income of 20,000 BDT (USD $237) and above had 3.4 times the odds (AOR 3.4, 95% CI 1.4–8.6) of prior knowledge about ANC services. Indigenous women who were aware of the nearest health care facilities had 4.3 times the odds of prior knowledge about ANC services (AOR 4.3, 95%CI 2.1–8.8). *Prior knowledge* of pregnancy complications was statistically significantly associated with prior knowledge of ANC services (AOR 3.6, 95% CI 1.6–8.1), however, *experiencing* pregnancy-related complications was not independently associated with having prior ANC knowledge. See Table 3.

### Factors associated with attending ANC services during pregnancy

Characteristics associated with ANC attendance for 232 Indigenous women who reported attending at least one ANC service during their last pregnancy compared to Indigenous women who did not attend an ANC service are shown in Table 4. Those factors with a significant relationship on univariate analysis were included in the multivariable logistic regression analysis.

In the multivariable logistic regression analysis, four factors were independently associated with ANC service attendance. Indigenous women who resided at Khagrachhari Sadar upazila

**Table 2.** Socio-demographic characteristics associated with Antenatal Care (ANC) knowledge for Indigenous women cohort who participated in the survey; n = 438.

| Variable | Participants who knew about ANC services during last pregnancy n = 304 | | Participants who did not know about ANC services during last pregnancy n = 134 | | Pearson's Chi-Square test | | |
|---|---|---|---|---|---|---|---|
| | n | % † | n | % † | $\chi^2$ | df | p-value |
| **Place of residence** | | | | | | | |
| Matiranga | 210 | 69 | 115 | 86 | | | |
| Khagrachhari Sadar | 94 | 31 | 19 | 14 | 13.6 | 1 | **<0.001** |
| **Age (years)** | | | | | | | |
| 16–24 | 129 | 42 | 86 | 64 | | | |
| 25–29 | 111 | 37 | 35 | 26 | | | |
| ≥ 30 | 64 | 21 | 13 | 10 | 18.8 | 2 | **<0.001** |
| **Ethnicity** | | | | | | | |
| Chakma | 153 | 50 | 67 | 50 | | | |
| Marma | 78 | 26 | 22 | 16 | | | |
| Tripura | 73 | 24 | 45 | 34 | 6.6 | 2 | **0.04** |
| **Religion** | | | | | | | |
| Buddhist | 231 | 76 | 89 | 66 | | | |
| Sonatoni/Hindu | 73 | 24 | 45 | 34 | 4.3 | 1 | **0.04** |
| **Participants' school attendance** | | | | | | | |
| Did not attend–Primary | 104 | 34 | 101 | 75 | | | |
| Junior–Secondary | 137 | 45 | 31 | 23 | | | |
| Higher Secondary & above | 63 | 21 | 2 | 2.0 | 68.5 | 2 | **<0.001** |
| **School attendance of partners** | | | | | | | |
| Did not attend–Primary | 109 | 36 | 86 | 64 | | | |
| Junior–Secondary | 123 | 40 | 46 | 34 | | | |
| Higher Secondary & above | 72 | 24 | 2 | 2.0 | 44.8 | 2 | **<0.001** |
| **Occupation** | | | | | | | |
| Complete housewife | 158 | 52 | 56 | 42 | | | |
| Income generative works | 146 | 48 | 78 | 58 | 3.8 | 1 | **0.05** |
| **Occupation of partner** | | | | | | | |
| Daily labour/Farmer | 136 | 45 | 85 | 63 | | | |
| Service holder/business | 168 | 55 | 49 | 37 | 13.0 | 1 | **<0.001** |
| **Monthly household income (BDT)** | | | | | | | |
| 4000–9000 BDT | 106 | 35 | 92 | 69 | | | |
| 10000–19000 BDT | 80 | 26 | 31 | 23 | | | |
| 20000 & above BDT | 118 | 39 | 11 | 8 | 53.4 | 2 | **<0.001** |
| **Knowledge about any nearest facility** | | | | | | | |
| Yes | 279 | 92 | 91 | 68 | | | |
| No | 25 | 8.0 | 43 | 32 | 40.4 | 1 | **<0.001** |
| **Knowledge of type of nearest health care facilities** | | | | | | | |
| Did not know any | 25 | 8.0 | 43 | 32 | | | |
| Community-based facility | 97 | 32 | 58 | 43 | | | |
| Government hospitals | 77 | 25 | 29 | 22 | | | |
| Private facilities | 105 | 3 | 4 | 3.0 | 75.3 | 3 | **<0.001** |
| **Distance between health centre and *paras*** | | | | | | | |
| ≤5 km | 70 | 23 | 27 | 20 | | | |
| 6–10 | 164 | 54 | 65 | 49 | | | |

(*Continued*)

**Table 2.** (Continued)

| Variable | Participants who knew about ANC services during last pregnancy n = 304 | | Participants who did not know about ANC services during last pregnancy n = 134 | | Pearson's Chi-Square test | | |
|---|---|---|---|---|---|---|---|
| >10 km | 70 | 23 | 42 | 31 | 3.4 | 2 | 0.2 |
| **Accessed any media for MHC information** | | | | | | | |
| Yes | 224 | 74 | 128 | 96 | | | |
| No | 80 | 26 | 6 | 4 | 28.1 | 1 | **<0.001** |
| **Age during 1st pregnancy** | | | | | | | |
| Became pregnant < 20 years | 179 | 59 | 105 | 78 | | | |
| Became pregnant ≥20 years | 125 | 41 | 29 | 22 | 15.5 | 1 | **<0.001** |
| **Negative pregnancy outcome** | | | | | | | |
| Yes | 227 | 75 | 98 | 73 | | | |
| No | 77 | 25 | 36 | 27 | 0.11 | 1 | 0.735 |
| **Number of total pregnancies** | | | | | | | |
| 1–2 pregnancies | 220 | 72 | 90 | 67 | | | |
| 3 & above | 84 | 28 | 44 | 33 | 1.2 | 1 | 0.30 |
| **Previous knowledge about pregnancy-complications** | | | | | | | |
| Yes | 201 | 66 | 122 | 91 | | | |
| No | 103 | 34 | 12 | 9.0 | 29.8 | 1 | **<0.001** |
| **Pregnancy complications during last pregnancy** | | | | | | | |
| No problems reported | 99 | 33 | 64 | 48 | | | |
| Mild | 107 | 35 | 36 | 27 | | | |
| Moderate (debilitated daily life) | 11 | 34.0 | 6 | 4.0 | | | |
| Severe (need immediate medical attention) | 87 | 29 | 28 | 21 | 10.0 | 3 | **0.02** |

† = Column percentage; BDT = Bangladeshi Taka; df = degrees of freedom.

had 6.5 times the odds of attending ANC services compared to those who resided in Matiranga upazila (AOR 6.5, 95% CI 1.6–25). Indigenous women with a monthly household income of 20,000 BDT (USD $237) and above had 2.8 times the odds of attending ANC service compared to Indigenous women with a monthly household income between 4000 and 9000 BDT (USD $47.6-$107) (AOR 2.8, 95% CI 1.1–7.4). Indigenous women with knowledge of ANC benefits during pregnancy had nearly eight times the odds of attending ANC services compared to Indigenous women unaware of these benefits (AOR 7.7, 95% CI 3.6–16). Indigenous women who resided more than 10 kilometres from the nearest service had 68% reduced odds of attending ANC services (AOR 0.32, 95%CI 0.09, 1.1). See Table 5.

## Discussion

This study aimed to estimate the prevalence of Indigenous women from CHT Bangladesh accessing ANC services and to determine factors associated with ANC service knowledge and attendance. The study included 438 Indigenous CHT women from three different ethnic communities (50% Chakma, 23% Marma and 27% Tripura) in proportions generally representative of the ethnic distribution of the population in CHT [31]. Two-thirds of these women reported having prior knowledge of ANC services and the prevalence of attending ANC services was 53% during their last pregnancy. Prior knowledge of, and attendance at, ANC services differed among the three Indigenous community groups. Key modifiable factors associated with prior knowledge of ANC services included education and having knowledge

**Table 3. Univariate and final multivariable logistic regression model for identifying factors associated with having Antenatal Care (ANC) knowledge among Indigenous women in Chittagong Hill Tracts, Bangladesh, after adjusting for clustering by *para*; n = 438.**

| Variables | Number of events | Univariate analysis | | Multivariable analysis* | |
|---|---|---|---|---|---|
| | | OR | 95% CI | Adjusted OR | 95% CI |
| **Age** | | | | | |
| Young (16–29 years) | 328 | Ref | | Ref | |
| Adult (30 & above years) | 110 | **2.6** | **1.5–4.4** | **2.2** | **1.1–4.6** |
| **School attendance** | | | | | |
| Didn't attend-Junior | 281 | Ref | | Ref | |
| Secondary and above | 157 | **10** | **5.3–20** | **4.7** | **2.0–11** |
| **Monthly household income** | | | | | |
| 4000–9000 BDT | 198 | Ref | | Ref | |
| 10000–19000 BDT | 111 | **2.2** | **1.3–3.7** | 1.7 | 1.0–3.3 |
| 20000 & above BDT | 129 | **9.3** | **4.7–18** | **3.4** | **1.4–8.6** |
| **Know nearest facility** | | | | | |
| No | 68 | Ref | | Ref | |
| Yes | 370 | **5.3** | **3.1–9.1** | **4.3** | **2.1–8.8** |
| **Knowledge of pregnancy-related complications** | | | | | |
| No | 323 | Ref | | Ref | |
| Yes | 115 | **5.2** | **2.7–9.9** | **3.6** | **1.6–8.1** |
| **Pregnancy complications experienced according to severity** | | | | | |
| No problems reported | 163 | Ref | | Ref | |
| Mild | 143 | **1.9** | **1.2–3.1** | 1.3 | 0.70–2.6 |
| Moderate | 17 | 1.2 | 0.42–3.4 | 0.60 | 0.13–2.5 |
| Severe | 115 | **2.0** | **1.2–3.4** | 1.9 | 0.92–3.8 |

*Adjusted for all other variables in the model; OR = Odds ratio; AOR = Adjusted Odd Ratio; CI = Confidence Interval, BDT = Bangladeshi Taka.

about the nearest health facilities and pregnancy-related complications. The key modifiable factor associated with accessing ANC services was having prior knowledge about the benefits of ANC services.

Previously published research on ANC service utilization by Mru Indigenous women in the CHT was only 11.2% in 2011 [17]. Results of this current study from three prominent CHT Indigenous groups suggest that Indigenous women's access to ANC may be improving. However, other improvements in care delivery may have been in place in comparison to other areas not purposively selected. This information may assist in the development of policy and programs to address gaps in the health rights of underprivileged groups like Indigenous women in Bangladesh.

A systematic review in developing countries revealed that women with knowledge about ANC services were more likely to access ANC services during pregnancy [36], however, there has been a lack of evidence about Indigenous women's knowledge of ANC. Findings from this study revealed that age, participants' level of education, household income, prior knowledge of nearest health facilities, and knowledge about pregnancy-related complications during last pregnancy were important contributing factors to having knowledge about ANC for Indigenous women in CHT.

One third (31%; n = 134) of CHT Indigenous women reported not knowing about ANC services, indicating that Indigenous maternal health issues in the CHT are lagging despite overall progress of ANC coverage (74%) in Bangladesh [37]. Specific knowledge about available beneficial health programs and quality health services positively influences care-seeking

**Table 4. Socio-demographic characteristics associated with Antenatal Care (ANC) attendance for 304 Indigenous women cohort who reported having knowledge on ANC services during their last pregnancy.**

| Variable | Participants attended ANC services during last pregnancy n = 232 | | Participants did not attend ANC services during last pregnancy n = 72 | | Pearson's Chi-Square test ($\chi^2$) | | |
|---|---|---|---|---|---|---|---|
| | n | % † | n | % † | $\chi^2$ | Df | p-value |
| **Place of residence** | | | | | | | |
| Matiranga | 144 | 62 | 66 | 92 | | | |
| Khagrachhari Sadar | 88 | 38 | 6 | 8.0 | 22.5 | 1 | **<0.001** |
| **Age (years)** | | | | | | | |
| 16–24 | 88 | 38 | 41 | 57 | | | |
| 25–29 | 91 | 39 | 20 | 28 | | | |
| ≥ 30 | 53 | 23 | 11 | 15 | 8.1 | 2 | **0.01** |
| **Ethnicity** | | | | | | | |
| Chakma | 118 | 51 | 35 | 49 | | | |
| Marma | 65 | 28 | 13 | 18 | | | |
| Tripura | 49 | 21 | 24 | 33 | 5.6 | 2 | 0.06 |
| **Religion** | | | | | | | |
| Buddhist | 183 | 79 | 48 | 67 | | | |
| Sonatoni/Hindu | 49 | 21 | 24 | 33 | 4.5 | 2 | **0.03** |
| **Participants' school attendance** | | | | | | | |
| Did not attend–Primary | 68 | 29 | 36 | 50 | | | |
| Junior–Secondary | 102 | 44 | 35 | 49 | | | |
| Higher Secondary & above | 62 | 27 | 1 | 1.0 | 24.2 | 2 | **<0.001** |
| **School attendance of partners** | | | | | | | |
| Did not attend–Primary | 69 | 30 | 40 | 56 | | | |
| Junior–Secondary | 96 | 41 | 27 | 37 | | | |
| Higher Secondary & above | 67 | 29 | 5 | 7.0 | 21.6 | 2 | **<0.001** |
| **Occupation** | | | | | | | |
| Complete housewife | 131 | 57 | 27 | 37 | | | |
| Income generative works | 101 | 43 | 45 | 63 | 7.9 | 1 | **0.005** |
| **Occupation of partner** | | | | | | | |
| Daily labour/Farmer | 95 | 41 | 41 | 57 | | | |
| Service holder/ Business | 137 | 59 | 31 | 43 | 5.7 | 1 | 0.02 |
| **Monthly household income (BDT)** | | | | | | | |
| 4000–9000 BDT | 66 | 28 | 40 | 56 | | | |
| 10000–19000 BDT | 58 | 25 | 22 | 30 | | | |
| 20000 & above BDT | 108 | 47 | 10 | 14 | 27.3 | 2 | **<0.001** |
| **Knowledge of nearest health care facilities** | | | | | | | |
| Did not know any | 15 | 7.0 | 10 | 14 | | | |
| Community-based facility | 63 | 27 | 34 | 47 | | | |
| Government hospitals | 56 | 24 | 21 | 29 | | | |
| Private clinic/facilities | 98 | 42. | 7 | 10 | 28.0 | 3 | **<0.001** |
| **Distance between health centre and *paras*** | | | | | | | |
| ≤5 km | 59 | 25 | 11 | 15 | | | |
| 6–10 | 132 | 54 | 32 | 45 | | | |
| >10 km | 41 | 18 | 29 | 40 | 16.2 | 2 | **<0.001** |
| **Age during 1ˢᵗ pregnancy** | | | | | | | |
| Became pregnant < 20 years | 120 | 52 | 59 | 82 | | | |

*(Continued)*

**Table 4.** (*Continued*)

| Variable | Participants attended ANC services during last pregnancy n = 232 | | Participants did not attend ANC services during last pregnancy n = 72 | | Pearson's Chi-Square test ($\chi^2$) | | |
|---|---|---|---|---|---|---|---|
| Became pregnant ≥20 years | 112 | 48 | 13 | 18 | 20.7 | 1 | **<0.001** |
| **Any media access for MHC information** | | | | | | | |
| Yes | 75 | 32 | 5 | 7.0 | | | |
| No | 157 | 68 | 67 | 93 | 18.2 | 1 | **<0.001** |
| **If attending ANC service provide any benefits** | | | | | | | |
| Yes | 196 | 85 | 33 | 46 | | | |
| No/ Do not know | 36 | 15 | 39 | 54 | 44.2 | 1 | **<0.001** |
| **Negative pregnancy outcome** | | | | | | | |
| Yes | 49 | 21 | 28 | 39 | | | |
| No | 183 | 79 | 44 | 61 | 9.2 | 1 | **0.002** |
| **Number of total pregnancies** | | | | | | | |
| 1–2 pregnancies | 177 | 76 | 43 | 60 | | | |
| 3 & above pregnancies | 55 | 24 | 29 | 40 | 7.5 | 1 | **0.006** |
| **Knowledge of pregnancy-related complications** | | | | | | | |
| Yes | 82 | 35 | 18 | 25 | | | |
| No | 150 | 65 | 54 | 75 | 3.3 | 1 | 0.068 |
| **Pregnancy complications during last pregnancy** | | | | | | | |
| No complications | 74 | 32 | 25 | 35 | | | |
| Mild | 82 | 35 | 25 | 35 | | | |
| Moderate | 10 | 4.0 | **1** | 1.0 | | | |
| Severe | 66 | 25 | 21 | 29 | 1.4 | 3 | 0.695 |

† = Column percentage; df = degrees of freedom; BDT = Bangladeshi Taka.

**Table 5. Univariate and multivariate logistic regression model for identifying factors associated with attending Antenatal Care (ANC) services among Indigenous women in Chittagong Hill Tracts, Bangladesh, after adjusting for clustering by *para*; n = 304.**

| Variables | Number of events | Univariate analysis | | Multivariable analysis | |
|---|---|---|---|---|---|
| | | **OR** | **95% CI** | **Adjusted OR** | **95% CI** |
| **Place of residence** | | | | | |
| Matiranga | 210 | Ref | | Ref | |
| Khagrachhari Sadar | 94 | **6.7** | **2.8–16** | **6.5** | **1.6–25** |
| **Monthly household income (BDT)** | | | | | |
| 4000–9000 BDT | 106 | Ref | | Ref | |
| 10000–19000 BDt | 80 | 1.6 | 0.80–3.0 | 1.0 | 0.41–2.0 |
| 20000 & above BDT | 118 | **6.5** | **3.1–14** | **2.8** | **1.1–7.4** |
| **Distance to the nearest facility** | | | | | |
| ≤5 km | 70 | Ref | | Ref | |
| 6–10 | 164 | 0.80 | 0.40–1.6 | 0.76 | 0.25–2.3 |
| >10 km | 70 | **0.30** | **0.10–0.60** | 0.32 | 0.09–1.1 |
| **Knowledge of ANC benefits** | | | | | |
| No | 75 | Ref | | Ref | |
| Yes | 229 | **6.4** | **3.4–11** | **7.7** | **3.6–16** |

OR = Odds ratio; AOR = Adjusted Odd Ratio; CI = Confidence Interval, BDT = Bangladeshi Taka.

for MHC services [38, 39]. In Vietnam, women from ethnic minority groups and hilly remote areas had limited opportunities to access health education, and health services were of poor quality, contributing to their limited knowledge about ANC services [39]. Women with better economic status and with prior knowledge of pregnancy-related complications were highly likely to be aware of ANC services [39]. Similar findings were found in this study with Indigenous women in CHT with prior knowledge of pregnancy-related complications and higher income having approximately three times the odds of being aware and of attending ANC.

Previous studies in China and Vietnam, indicated that Indigenous women were not aware of existing specialised MHC services within their communities, and this information gap contributed to lower utilization of ANC services during pregnancy [25, 40]. Although there were no Indigenous-women focused MHC programs in Khagrachhari district, there were low-cost public facilities where ANC services were available [31, 41]. Findings indicated a knowledge gap about existing nearby low-cost public facilities for attending ANC services [31]. This is a clear indication of an information and communication gap between service providers and service receivers contributing to health inequality and inequity [28, 42]. To improve health outcomes, Indigenous people should be aware of existing primary health care services and those services should be available, affordable and acceptable for the community [43].

In this study Indigenous women with higher education were nearly five times more likely to know about ANC services compared to those with lower levels of education. Previous research suggests educated women have more knowledge about self-care [38, 44]. Results suggest that Indigenous women who knew about their nearest health facilities and knew about pregnancy-related complications were more likely to be aware of ANC services. Previous findings suggest that contact with health care providers increases knowledge about maternal health issues and motivates pregnant women to access further services [38]. Half the CHT Indigenous women (55%, n = 168) received ANC related information from health care providers, and completed four ANC visits (54%, n = 49) during their last pregnancy. These findings suggest that knowledge about ANC services is essential to access.

Indigenous women in the CHT are subject to several socio-economic barriers to attendance of MHC services including greater distance to facilities and lower levels of education and income [11, 24, 27]. Distance to nearest facilities and household income were key factors for ANC attendance in this study supported by previous findings fromLMICs [24, 27, 45]. Indigenous women residing in Khagrachhari Sadar Upazila were more likely to access ANC services compared to those residing in Matiranga Upazila. District level hospitals, private clinics and NGO-based health facilities were located at Khagrachhari Sadar upazilla and not at Matiranga upazila which might contribute towards community awareness [31]. Better transport systems play a vital role in accessing health facilities more easily [31, 46]. In contrast to previous research however, level of education was not an independent factor for attending ANC services in this study [17, 23, 26, 35]. In terms of ANC attendance, this study revealed a previously undocumented factor associated with ANC attendance which is prior knowledge of ANC benefits.

Consistent with previous studies in Bangladesh, India and Vietnam ANC attendance was higher among Indigenous women from higher-income households in this study [17, 24, 27, 39]. Although a significant number of participants were from a lower socio-economic background, private ANC services were most comonly accessed. A significant proportion of Indigenous women accessed ANC services due to complications (22.4%) and the unadjusted odds of attending ANC services were statistically significantly higher for women with severe complications. Women from disadvantaged groups, including Indigenous women, access private facilities for quicker and better quality care [39] and when there are complications, even though cost may be an issue [38].

Like Indigenous women in India, ANC service attendance was lower among working Indigenous women in the CHT however after adjusting for other factors it did not remain independently associated with attendance [27]. Indigenous women in Khagrachhari were involved in income generation activities; however most had lower levels of education and worked as a labourer where priorities were more related to money and time rather than concentrating on their pregnancy which they perceived as less risky [25]. Working women are expected to have more freedom and more knowledge about pregnancy, and are therefore in theory more likely to access healthcare facilities [47].

Regarding obstetric history of Indigenous women in the CHT, similar to the Scheduled Tribes women in India, the majority conceived at a young age [26]. Experiencing pregnancy-related complications was statistically significantly associated with ANC attendance among ethnic minority groups from a resource-poor region of Sichuan Province in China [25], however, this was not identified as an independent factor among CHT Indigenous women.

Accessing medically trained health care providers (such as doctors, nurses, community-based skilled birth attendants) for ANC checks was lower among CHT Indigenous women compared to the national average (53% versus 64%) [34]. Doorstep services provided by community-based health workers can have a positive influence on Indigenous women accessing ANC services [24, 26]. Rates of accessing community-based health facilities and community-based health workers for ANC checks in this study are similar to accessing private facilities. Indigenous communities are expected to benefit from community-based facilities for primary health care needs [41]. Community-based health workers are expected to be the first point of contact for the community because they share common cultural values and life experiences with the people they serve [48, 49]. The Government of Bangladesh has adopted a community-based approach to address significant shortages of human resources in the health sector [50]. Therefore, roles of community-based health workers need to be examined to understand their influence or contribution to MHC services utilization among Indigenous communities.

Using Indigenous language by health care providers while providing health care contributes towards improving MHC service access during pregnancy and at childbirth among Indigenous women [51, 52]. Not having consultation in Indigenous languages might restrict opportunities to understand health information provided to them as well as affecting further service intake [17, 53].

Of those women who attended ANC services, most (63.4%) did not have their first ANC check within the first trimester; rather they accessed ANC services in the second trimester, mostly due to health problems [26]. According to recent national survey data, the prevalence of pregnant women attending at least four ANC-visits in Bangladesh is 37%, a similar percentage to Indigenous women in Khagrachhari district (39%) [37, 54]; however, this percentage varied from 21% to 51% among ethnic groups. A cross-sectional study conducted in four different states in India reported that attending the recommended 4-ANC-visits was lower among tribal women (ranging from 3.6% to 14%) suggesting that most Indigenous women did not have knowledge about the importance of attending [24].

Findings strongly suggest that data on ANC attendance needs to be monitored and public health interventions developed to assist women to improve access to ANC services [26]. Further research is also required to explore affordability and acceptability of existing ANC services in the CHT.

## Strengths and limitations

As an observational design, this cross-sectional study collected data at a single time point and is therefore unable to establish temporal sequence [55]. However, cross-sectional studies

provide preliminary insights into situations and can be helpful for health services research [55]. Although it is a snapshot of Indigenous women's health in the CHT, this is a pioneer study on Indigenous women's access to existing MHC services in the CHT. The uniqueness of this study is that it was conducted in the three dominant ethnic communities of the Khagrach-hari district, representing the major ethnic groups in the two other hill districts sharing similar socio-cultural status [13]. Data collection was designed to reduce bias and increase generalisability by attempting to recruit all eligible Indigenous women in each *para* of the study sites [56]. Not using Indigenous native terms for pregnancy-related complications may have restricted data collection from some participants [57]. Findings provide a reasonable estimate of prevalence for accessing ANC by Indigenous women and are generalizable within these CHT communities providing useful insights for optimising existing ANC service use for Indigenous communities.

## Conclusion

The estimated prevalence of accessing ANC services among Indigenous women in the CHT is suboptimal at around half. Lower utilization of ANC services among Indigenous women is associated with socio-demographic characteristics such as age, household income, distance to nearest primary health care facilities and knowledge of ANC benefits. Indigenous women in the CHT had limited knowledge of nearby health facilities and pregnancy-related complications. Given the importance of accessing ANC services to improve health outcomes for all, addressing these gaps in access is critical. Information on accessing ANC services among Indigenous women should be addressed to accelerate the increase of ANC utilization in Bangladesh, including using Indigenous native language. Engaging Indigenous women, family and ethnic communities in the design of health care interventions would be a first step in creating a sense of ownership and appropriateness of health services to ensure greater access and sustainability, ultimately improving maternal healthcare.

## Supporting information

**S1 Table. All data related to ANC service access for the manuscript "prevalence and factors associated with antenatal care service access among Indigenous women in the Chittagong Hill Tracts, Bangladesh: A cross-sectional study" (Akter et al.).**
(PDF)

**S2 Table. Variable dictionary for "prevalence and factors associated with antenatal care service access among Indigenous women in the Chittagong Hill Tracts, Bangladesh: A cross-sectional study" (Akter et al.).**
(PDF)

## Acknowledgments

The authors are indebted to all the Indigenous women who took part in the study and shared their information for this research, and local community leaders for their invaluable support. Special thanks to the field research assistants for their incredible work during data collection.

## Author Contributions

**Conceptualization:** Shahinoor Akter, Kerry Jill Inder.

**Data curation:** Shahinoor Akter.

**Formal analysis:** Shahinoor Akter, Kerry Jill Inder.

**Methodology:** Shahinoor Akter, Kerry Jill Inder.

**Project administration:** Shahinoor Akter.

**Supervision:** Jane Louise Rich, Kate Davies, Kerry Jill Inder.

**Validation:** Shahinoor Akter, Kerry Jill Inder.

**Visualization:** Shahinoor Akter.

**Writing – original draft:** Shahinoor Akter.

**Writing – review & editing:** Jane Louise Rich, Kate Davies, Kerry Jill Inder.

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
