## [Decision Letter · Decision Letter 0]

21 Apr 2020

PONE-D-19-33547

Prevalence and factors associated with antenatal care service access among Indigenous women in the Chittagong Hill Tracts, Bangladesh: A cross-sectional study

PLOS ONE

Dear Dr Akter,

Thank you for submitting your manuscript to PLOS ONE. After careful consideration, we feel that it has merit but does not fully meet PLOS ONE’s publication criteria as it currently stands. Therefore, we invite you to submit a revised version of the manuscript that addresses the points raised during the review process.

We would appreciate receiving your revised manuscript by 30/05/2020. To enhance the reproducibility of your results, we recommend that if applicable you deposit your laboratory protocols in protocols.io, where a protocol can be assigned its own identifier (DOI) such that it can be cited independently in the future. For instructions see: http://journals.plos.org/plosone/s/submission-guidelines#loc-laboratory-protocols

We look forward to receiving your revised manuscript.

Kind regards,

Yeetey Akpe Kwesi Enuameh, MD, MSc, DrPH

Academic Editor

PLOS ONE

Journal Requirements:

2. Please amend your current ethics statement to address the following concerns: Please explain why was written consent was not obtained, how you recorded/documented participant consent, and if the ethics committees/IRBs approved this consent procedure.

3. During our internal checks, the in-house editorial staff noted that you conducted research in another country. Please check the relevant national regulations and laws applying to foreign researchers and state whether you obtained the required approvals. Please address this in your ethics statement in both the manuscript and submission information. In particular, please state whether the study was approved by a Bangladeshi ethics committee.

You indicated that you had ethical approval for your study. In your Methods section, please ensure you have also stated whether you obtained consent from parents or guardians of the minors included in the study or whether the research ethics committee or IRB specifically waived the need for their consent.

4. Please include additional information regarding the survey or questionnaire used in the study and ensure that you have provided sufficient details that others could replicate the analyses. For instance, if you developed a questionnaire as part of this study and it is not under a copyright more restrictive than CC-BY, please include a copy, in both the original language and English, as Supporting Information.  If the original language is written in non-Latin characters, for example Amharic, Chinese, or Korean, please use a file format that ensures these characters are visible."

5. Please state whether you validated the questionnaire prior to testing on study participants. Please provide details regarding the validation group within the methods section.

6. We note that you have indicated that data from this study are available upon request. PLOS only allows data to be available upon request if there are legal or ethical restrictions on sharing data publicly. For information on unacceptable data access restrictions, please see http://journals.plos.org/plosone/s/data-availability#loc-unacceptable-data-access-restrictions.

Additional Editor Comments (if provided):

Please address comments from the reviewer and those following. Would advise you consult the Strobe checklist in shaping up your manuscript:

1. Rephrase line 61, it is not that clear. Also be consistent in your use of percentages or absolute numbers when presenting the results.

2. Lines 96, 116, 162, etc.: New sentences are usually not started with abbreviations like ANC- you start them with the full expression

3. Line 96 - 98: The sentence is not clear - revise

4. Line 114: Last sentence needs revision - possibly "after twenty years of signing the peace accord"

5. Lines 130 - 131: The part of the sentence beyond the "comma" is not clear - revise

6. Lines 136 - 137: The expression "the top down nature of health intervention programs" could mean different things to different readers, so please be more specific

7. Lines 146 - 149: Under "methods", provide information on the study setting including all the communities you mention in the results section that were not earlier mentioned - that could include info on health facilities, etc. Also, how did you arrive at the sample size??? Could you provide reasons for participants having to be "within 36 months of delivery"?

8. Lines 156 - 160: Would it not be better describing the data collection tool fully first, before describing the data collection approach?

9. Lines 163 - 168: The sentence is exceedingly long and quite difficult following... please revise

10. Line 177: The expression "prevalence of having knowledge" - though seen it in a write-up, sounds interesting, why not simply "knowledge of ...."???

11. Lines 179 - 183: Long sentence, revise to make it easy appreciating

12. Lines 191 - 192: I do not sure of the reason behind the use chi-squares and then odd-ratios when the latter could have equally addressed the relationships of interest.

13. Lines 194 - 196: could you please provide the reasons for the varying "p-value" thresh-holds mentioned in this section.

14. Lines 199: "factors of interest" to what? Seems to be hanging... Or they are the "independent variables" for which the associations will be generated?

15. Line 207: "Given small cell sizes, categories were collapsed to..." - Is there not the possibility that not everyone will understand the expressions 'small cell sizes" and "categories collapsed" used in this sentence? So please rephrase.

16. Line 216 and beyond: could be "ethical considerations" instead? Provide "ethical approval number". Describe how privacy and confidentiality was maintained.

17. Line 222 and beyond: The results section requires some reshaping

a. Would recommend you start with a brief introduction describing the response rate and some other relevant information.

b. Then provide detailed descriptive information on socio-demographic characteristics and any other relevant information in a table.

c. You could present information on prevalence - those could be in the same table as the socio-demographics and you could provide narratives to them.

d. You then work on the associations, with their tables for knowledge and access i.e. unadjusted and adjusted OR. I do not see the importance of Chi square here as the odds ratios much more address your objectives than the former.

e. Also with results of logistic regression, would it not be good to mention those that were significant in univariate analysis that were then moved into multivariate for analysis before reporting the outcome of the latter. f. Information in figures 1 & 2 could be captured either in the tables on socio-demographics or prevalence - that should tell the story better than solitary figures...

18. Lines 224 - 227: Not clear on the role of this information - what is the role of CHW and Kabari??? this was not mentioned in the methods section. Also were these part of your sample??? You have also introduced new terms here Matiranga, etc. which are not mentioned anywhere else...

19. Lines 280, etc.: Ages would better be categorized as "25 - 29" and "30 and above", unless there are other reasons why that would not be possible

20. Lines 355 - 359: As you mentioned earlier, you "purposively" selected communities where data was available to address the study objectives. Could it also not be that other improvements in care delivery were in place in comparison to other places you did not select?

Reviewers' comments:

Reviewer's Responses to Questions

**Comments to the Author**

1. Is the manuscript technically sound, and do the data support the conclusions?

Reviewer #1: Yes

2. Has the statistical analysis been performed appropriately and rigorously? 

Reviewer #1: Yes

3. Have the authors made all data underlying the findings in their manuscript fully available?

Reviewer #1: Yes

4. Is the manuscript presented in an intelligible fashion and written in standard English?

Reviewer #1: Yes

5. Review Comments to the Author

Reviewer #1: General Comment: This is a well written paper which sought to assess the prevalence and factors associated with knowledge and access of Antenatal Care Services among the indigenous women in Bangladesh. I think that appropriate study methods were used in the conduct of this study. However, the authors need to beef up the method section and verify some few portions of the results sections. These are minor revisions that the authors can easily address.

Study design and Participants

Though reference has been made to a previously published paper which has full details of the methodology, I think the manuscript should be self-explanatory. Thus, it would be helpful to provide a little more information on the methods. Particularly, there should be some information about the study setting, eligibility criteria, selection of participants/sampling and sample size estimation. I think authors should briefly explain these under appropriate headings under methods as this will help to situate the study results in context.

It is also not clear whether this was a facility or community-based survey?

Line 202 under the data analysis, expected number of children was stated as a variable of interest but this variable was neither described nor reported on. Why was it not reported?

Ethics: Did participants give written/verbal consent? Also, considering the age range of participants, can you explain how you handled the ethical issues of women who were under 18 years?

Results

Table 1: It appears outputs for the variables “Experienced pregnancy complications during last

pregnancy” (yes//no) and “Pregnancy complications experienced according to severity” have been interchanged. This is because 99 participants responded yes to the first question and thus the various forms of severity should add up to 99, however, it is the reverse. Kindly check again.

Table 4

Lines 211-213: How was knowledge on pregnancy-related complications categorized into mild, moderate and severe? It appears this should rather be the experience of pregnancy-related complications and not the knowledge of it. Kindly check on this.

Line 256: I am not too sure why you would cite this reference while you are presenting the study results. I think it should be taken off else it would appear as if you are already discussing the results.

Line 328-330: Please check the odds of indigenous women who reside more than 10 kilometers from the nearest facility again. It differs from what is reported in Table 4, (the AOR from Table 4 is 0.32 (95%CI 0.09-1.1).

Lines 414-416: Please clarify this sentence. With regards to the use of modifiable factors, I will suggest that you do away with it as captured in lines 346 and 348 to ensure consistency because right from the background no mention was made of key modifiable factors and besides, your focus is on factors, whether modifiable or not so why do you want to qualify it now.

Figure 1

Finally, I suggest the title of Figure 1 (i.e Distribution of Indigenous women by type of ANC service received) is changed to “Distribution of Indigenous women by facility attended. I see the private facilities, community-based facilities and secondary hospital shown in the figure to be types of health facilities and not ANC services provided. The ANC services have to do with the checking of the vitals (weight, BP, etc), the urine and HIV test and detection of pregnancy-related complications among others.

Again, the figure is confusing, why is the sum of lines 3-5 in Figure 1 more (317) than the total number of women who accessed ANC at least once (232)? I think lines 3-7 in Figure 1, should add up to the total number of indigenous women who participated in the study (i.e. 438) but that is not it.

6. PLOS authors have the option to publish the peer review history of their article (what does this mean?). If published, this will include your full peer review and any attached files.

Reviewer #1: Yes: Grace Manu

---

## [Author Response · Author response to Decision Letter 0]

19 Jun 2020

A rebuttal letter titled "Response to reviewer" has been uploaded addressing each comments received from the reviewers on 21 April 2020 along with the manuscript and on 03rd June 2020 sent via email from the in-house team.

---

## [Decision Letter · Decision Letter 1]

17 Sep 2020

PONE-D-19-33547R1

Prevalence and factors associated with antenatal care service access among Indigenous women in the Chittagong Hill Tracts, Bangladesh: A cross-sectional study

PLOS ONE

Dear Dr. Akter,

Thank you for submitting your manuscript to PLOS ONE. After careful consideration, we feel that it has merit but does not fully meet PLOS ONE’s publication criteria as it currently stands. Therefore, we invite you to submit a revised version of the manuscript that addresses the points raised during the review process.

Please address the reviewer and editor suggestions as stated.

We look forward to receiving your revised manuscript.

Kind regards,

Yeetey Akpe Kwesi Enuameh, MD, MSc, DrPH

Academic Editor

PLOS ONE

Additional Editor Comments (if provided):

The authors have significantly revised and improved the manuscript. A few suggestions here and further by reviewer 1 should be addressed.

1. Sample size was 421 in the methods section, but 494 in the results section. Please rectify that discrepancy.

Thank you.

Reviewers' comments:

Reviewer's Responses to Questions

**Comments to the Author**

1. If the authors have adequately addressed your comments raised in a previous round of review and you feel that this manuscript is now acceptable for publication, you may indicate that here to bypass the “Comments to the Author” section, enter your conflict of interest statement in the “Confidential to Editor” section, and submit your "Accept" recommendation.

Reviewer #1: (No Response)

2. Is the manuscript technically sound, and do the data support the conclusions?

Reviewer #1: (No Response)

3. Has the statistical analysis been performed appropriately and rigorously? 

Reviewer #1: (No Response)

4. Have the authors made all data underlying the findings in their manuscript fully available?

Reviewer #1: (No Response)

5. Is the manuscript presented in an intelligible fashion and written in standard English?

Reviewer #1: (No Response)

6. Review Comments to the Author

Reviewer #1: The authors have done well by revising the manuscript. Some few concerns have been noted in the revised version, and would have to be addressed before publishing this manuscript.

Abstract

1. Line 61: Since the 494 women here refers to only women who met the inclusion criteria and not the total number of women in the two sub districts, I suggest you qualify it as such. Thus, the sentence could read “Of 494 indigenous women who met the inclusion criteria …… 438 participated (89% response rate) in the study”.

2. Lines 70 and 395 are not consistent: While line 70 states that “Indigenous women residing in Matiranga district (OR 6.5, 95%CI 1.7-25)”, line 395 on the other hand has it that “Indigenous women who resided at Khagrachhari Sadar sub district had 6.5 times the odds of attending ANC services”.

Introduction

1. Line 85: I suggest you rephrase the sentence as “A significant increase in the use of maternal health care (MHC) services …….”, for a better flow.

2. Line 97-99: This portion of the sentence “… including about other MHC services including safe delivery ….” needs to be revised.

3. Line 140-142: There seem to be some omission in the sentence “…. where community- engagement was not priorotised resulted in culturally unfriendly for MHC services.”

Study Setting

1. Line 161-165: I think this will better fit under the section on data sources in page 8, and please indicate if the data collection tool was pre-tested and the people among whom it was done.

2. Again under the study setting, I wish you provide some information on maternal health care services available in the area.

3. Line 162: Since the individual women who participated in the study did not represent their households, I suggest the sentence is revised to read “The survey was conducted in three communities” to do away with the households.

Variables

1. Lines 175-181: Under this section, the authors just stated the variables without indicating the outcome and independent variables. I suggest this section is reorganized to incorporate the outcome variables and lines 261-269 (under the statistical methods), under a common heading.

2. Line 261-263: “Where varibale categories were were less than 5 for participants’ and their partners’ education status, and for participants’ and partners’ occupation status repsonses were combined to allow statistical regression”. This sentence has to be revised to clarify and correct the typographical errors.

• Do you mean to say that variables for education and occupation of participants and their partners had more than 5 categories and were thus categorized or the variables rather had less categories?

• How was the categorization done or which variables were combined?

3. Line 351: Please move the “yes/no” to the variables section or an appropriate section where the various groupings of each variable is described.

Results

1. Is line 353 “These variables, significant on univariate analysis were included in the multivariable logistic” implying that the variables stated in lines 346-354 were significant in the univariate analysis?

• Why was the univariate result not presented instead of the chi-square test outputs?

2. Line 387: Same concern as above; why not present the univariate outputs as the multivariate analysis was dependent on results from the univariate analysis and not the chi-square test?

3. Please, be consistent in your report of the categories of the age (years) variable as the categories presented in the tables (16-24 years) is different from what is reported in lines 62 and 301-302 (15-29 years).

4. Lines 373-374 and Lines 439-440 seem to be contradictory. While lines 373-374 reports that “… experiencing pregnancy-related complications was not independently associated with having prior ANC knowledge”, lines 439-440 reports that “…. experiencing pregnancy-related complications …. was an important contributing factor to having knowledge about ANC…”, please reconcile these.

7. PLOS authors have the option to publish the peer review history of their article (what does this mean?). If published, this will include your full peer review and any attached files.

Reviewer #1: **Yes: **Grace Manu

---

## [Author Response · Author response to Decision Letter 1]

8 Oct 2020

A separate document labelled as "Response to Reviewer PONE-D-19-33547" has been uploaded.

---

## [Editor Report · Decision Letter 2]

15 Dec 2020

Prevalence and factors associated with antenatal care service access among Indigenous women in the Chittagong Hill Tracts, Bangladesh: A cross-sectional study

PONE-D-19-33547R2

Dear Dr. Shahinoor Akter,

We’re pleased to inform you that your manuscript has been judged scientifically suitable for publication and will be formally accepted for publication once it meets all outstanding technical requirements.

Kind regards,

Yeetey Akpe Kwesi Enuameh, MD, MSc, DrPH

Academic Editor

PLOS ONE

Additional Editor Comments (optional):

Comments addressed but a proof read ahead of publication will be helpful
---

## [Editor Report · Acceptance letter]

17 Dec 2020

PONE-D-19-33547R2 

Prevalence and factors associated with antenatal care service access among Indigenous women in the Chittagong Hill Tracts, Bangladesh: A cross-sectional study 

Dear Dr. Akter:

I'm pleased to inform you that your manuscript has been deemed suitable for publication in PLOS ONE. Congratulations! Your manuscript is now with our production department. 

Kind regards, 

on behalf of

Dr. PLOS Manuscript Reassignment 

Staff Editor

PLOS ONE